# C-reactive protein and risk of cognitive decline: The REGARDS study

**Miguel Arce Rentería**[1], Sarah R. Gillett[2], Leslie A. McClure[3], Virginia G. Wadley[4], Stephen P. Glasser[4], Virginia J. Howard[5], Brett M. Kissela[6], Frederick W. Unverzagt[7], Nancy S. Jenny[8†], Jennifer J. Manly[1], Mary Cushman[9]*

**1** Department of Neurology, Taub Institute for Research on Alzheimer's Disease and the Aging Brain, Columbia University College of Physicians and Surgeons, New York, New York, United States of America, **2** Department of Medicine, Larner College of Medicine at the University of Vermont, Burlington, Vermont, United States of America, **3** Departments of Epidemiology and Biostatistics, Drexel University, Philadelphia, Pennsylvania, United States of America, **4** Department of Medicine, University of Alabama at Birmingham, Birmingham, Alabama, United States of America, **5** Department of Epidemiology, University of Alabama at Birmingham, Birmingham, Alabama, United States of America, **6** Department of Neurology and Rehabilitation Medicine at the University of Cincinnati, Cincinnati, Ohio, United States of America, **7** Department of Psychiatry, Indiana University School of Medicine, Indianapolis, Indiana, United States of America, **8** Department of Pathology & Laboratory Medicine, Larner College of Medicine at the University of Vermont, Burlington, Vermont, United States of America, **9** Departments of Medicine and Pathology & Laboratory Medicine, Larner College of Medicine at the University of Vermont, Burlington, Vermont, United States of America

† Deceased.
* mary.cushman@uvm.edu

**Data Availability Statement:** The data underlying the findings include potentially identifying participant information, and cannot be made publicly available due to ethical/legal restrictions. However, data including statistical code from this

## Abstract

Markers of systemic inflammation are associated with increased risk of cognitive impairment, but it is unclear if they are associated with a faster rate of cognitive decline and whether this relationship differs by race. Our objective was to examine the association of baseline C-reaction protein (CRP) with cognitive decline among a large racially diverse cohort of older adults. Participants included 21,782 adults aged 45 and older (36% were Black, *Mean* age at baseline 64) from the REasons for Geographic And Racial Differences in Stroke (REGARDS) study. CRP was measured at baseline and used as a continuous variable or a dichotomous grouping based on race-specific 90th percentile cutoffs. Cognitive measures of memory and verbal fluency were administered every 2 years for up to 12 years. Latent growth curve models evaluated the association of CRP on cognitive trajectories, adjusting for relevant demographic and health factors. We found that higher CRP was associated with worse memory (B = -.039, 95% CI [-.065,-.014]) and verbal fluency at baseline (B = -.195, 95% CI [-.219,-.170]), but not with rate of cognitive decline. After covariate adjustment, the association of CRP on memory was attenuated (B = -.005, 95% CI [-.031,-.021]). The association with verbal fluency at baseline, but not over time, remained (B = -.042, 95% CI [-.067,-.017]). Race did not modify the association between CRP and cognition. Findings suggest that levels of CRP at age 45+, are a marker of cognitive impairment but may not be suitable for risk prediction for cognitive decline.

manuscript are available to researchers who meet the criteria for access to confidential data. Data can be obtained upon request through the University of Alabama at Birmingham at regardsadmin@uab. edu.

**Funding:** This research project is supported by cooperative agreement U01 NS041588 co-funded by the National Institute of Neurological Disorders and Stroke (NINDS) and the National Institute on Aging (NIA), National Institutes of Health, Department of Health and Human Service. The content is solely the responsibility of the authors and does not necessarily represent the official views of the NINDS or the NIA. Representatives of the NINDS were involved in the review of the manuscript but were not directly involved in the collection, management, analysis or interpretation of the data. Additional funding was provided by National Heart Lung and Blood Institute T32 HL07594-24 and National Institute of General Medical Sciences P20 GM135007.

**Competing interests:** The authors have declared that no competing interests exist.

# 1. Introduction

As global population aging increases, the prevalence of cognitive impairment and neurocognitive disorders, such as Alzheimer's disease and related disorders, will increase dramatically [1, 2]. Higher prevalence of cognitive impairment among aging adults is a public health concern, as it is associated with increased rates of disability [3], larger health care costs [4], and increase risk of dementia [5]. As such, identification of those at highest risk for cognitive decline might allow targeted prevention efforts.

Inflammation may be an important mechanism underlying risk for cognitive impairment and dementia [6–8]. C-reactive protein (CRP) is a marker of acute inflammation in acute illness, but low level inflammation in healthy people is captured by high sensitivity assays and is related to a variety of disease outcomes [9]. A number of studies suggest that CRP might be associated with cognitive impairment [10–13], with some evidence of increased risk of cognitive decline [14–16]. While some prospective studies have found higher rates of cognitive decline among individuals with higher CRP [17–19], the majority of these studies were among highly selected clinic or cohort samples of adults. In order to generalize the relationship between CRP and cognitive outcomes, a national, longitudinal, population-based sample of adults is needed.

Associations between CRP and cognitive decline may be moderated by race. Generally, Black Americans are at higher risk of cognitive impairment compared to White Americans [20, 21]. Similarly, they have higher levels of CRP than their White counterparts [22–24]. There is some evidence to suggest that Black people may respond to inflammatory stimuli differently than White people [25, 26]. Gene variants that up-regulate proinflammatory cytokines are also more common in Black than White Americans [26]. Given these findings, it can be hypothesized that Black Americans may be at higher risk for inflammation-related cognitive decline.

The objective of this study was to examine the association between CRP and cognitive trajectories in a national, population-based cohort of Black and White American adults. We hypothesized that higher baseline CRP concentration would be associated with worse cognitive functioning at baseline and a steeper rate of cognitive decline over time, independent of other risk factors. Additionally, we evaluated whether any association of CRP with cognition would be moderated by race such that the association would be stronger among Black compared to White people.

# 2. Methods

## 2.1. Design and procedures

The REasons for Geographical and Racial Differences in Stroke (REGARDS) study is a national, population-based prospective cohort study of Black and White Americans aged ≥45 years at baseline [27]. The cohort includes 30,239 participants, 45% men and 55% women, 58% White and 42% Black, 56% residing in the southeastern Stroke Belt region of United States (Alabama, Arkansas, Georgia, Louisiana, Mississippi, North Carolina, South Carolina, and Tennessee) and 44% in the remaining 40 contiguous United States. The Stroke Belt region has well-documented higher rates of stroke than the rest of the United States [28]. Participants were recruited from commercially available lists of U.S. residents using mail and telephone contact. Enrollment occurred between January 2003 and October 2007. Interviewers were trained to identify participants answering questions in a manner suggesting lack of comprehension, and such participants were not included further. Baseline demographic information, medical history, and health status were collected by computer-assisted telephone interview (CATI) with follow-ups occurring every 6 months (maximum follow-up up to 12 years). Trained health care professionals collected blood and urine samples, electrocardiogram, blood pressure, height, and weight

during an in-home visit at baseline. Further methodological details are available elsewhere [27], but in brief, blood pressure quality control was monitored by central examination of digit preference, height was measured once utilizing an 8-foot metal tape measure and a square, and weight (without shoes) was measured once using a standard 300-lb calibrated scale.

As shown in Fig 1, for the current project, participants were excluded if they reported a history of stroke at baseline, cognitive impairment at baseline based on the Six Item Screener (SIS score ≤4) [29], or if missing CRP data. The resulting sample size for analysis was 21,782.

REGARDS was approved by Institutional Review Boards of all participating institutions and all participants provided written informed consent. Potential participants who were able to respond to telephone questions provided verbal consent, which was followed by written consent at an in-home visit. The current study was approved by the Institutional Review Boards of Columbia University Medical Center and Larner College of Medicine at the University of Vermont.

## 2.2. Inflammatory biomarkers and laboratory analysis

At the in-home visit, blood was collected by trained personnel using standardized procedures after a 10–12 hour fast and centrifuged within 2 hours of collection. Plasma and serum were separated and shipped overnight on gel ice packs to a central laboratory. Samples were re-centrifuged and stored for batch processing [27]. Lipid profile and glucose were measured using the Ortho Vitros Clinical Chemistry System 950IRC instrument. CRP was measured in plasma with a high-sensitivity, particle enhanced immunonephelometric assay (N High Sensitivity CRP, Dade Behring Inc., Deerfield, IL; interassay CVs 2.1–5.7%). Validity of results using this blood collection method was confirmed using a paired samples technique [30] and the assay has reasonable within person variability [31]. In addition to evaluating CRP as a continuous

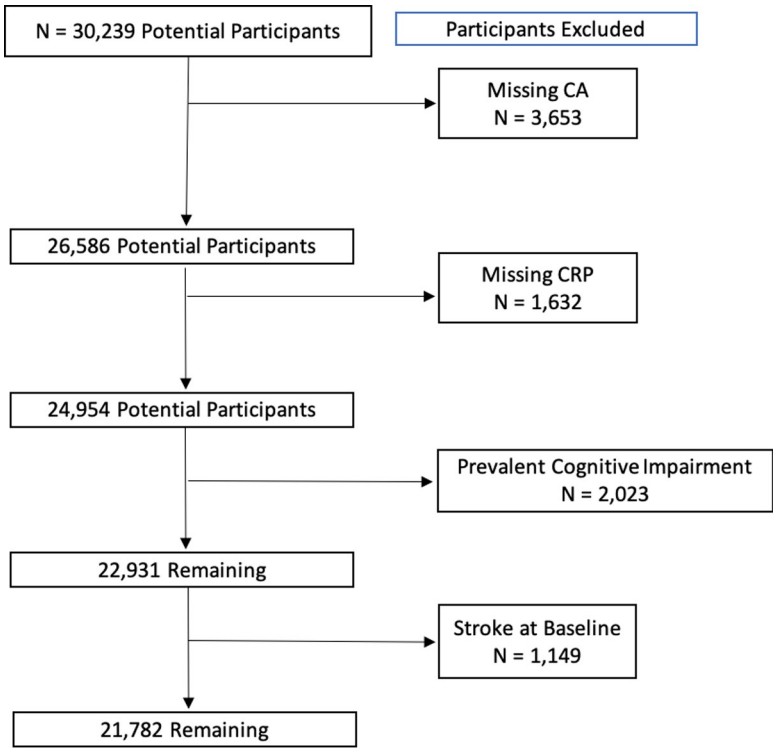

**Fig 1. Flow diagram for identifying participants.** CA = Cognitive Assessment.

variable (with log transformation to correct for skewness), all participants were categorized as having either having CRP "above" or "below" race-specific $90^{th}$ percentile cutoffs ($\geq 8$ mg/L for White adults and $\geq 12.3$ mg/L for Black adults). Although studies suggest that a CRP $\geq 3$ mg/L indicates levels of inflammation important to disease risk prediction [32, 33], we chose a race-specific higher cutoff for elevated CRP because differences in CRP are partially driven by genetic ancestry, and those with African ancestry have higher values than all other groups, suggesting that a single threshold value may not be appropriate [22]. CRP $\geq 3$ mg/L was also not associated with risk of future stroke in Black Americans [34]. We chose race-specific $90^{th}$ percentile in order to select the highest possible range of CRP for given racial group.

## 2.3. Cognitive assessment

Starting in 2006 and repeated every two years, participants completed cognitive measures over the phone administered by trained interviewers [35]. The cognitive measures assessed the cognitive domains of memory and verbal fluency/executive functioning. Episodic memory was assessed through the Consortium to Establish a Registry for Alzheimer's Disease (CERAD) Word List [36]. Participants are asked to recall a list of 10 unrelated words across three learning trials, and after a 5-minute delay, they are asked to recall as many words as possible from the list. Total words recalled across the three learning trials were summed to create an immediate recall score. The immediate and delayed recall scores were converted to z-scores using the entire sample's means and standard deviation at the initial cognitive assessment. An episodic memory composite score was derived as the average of the immediate and delayed recalled z-scores at each visit. Verbal fluency and executive functioning were assessed with tests of letter and semantic fluency. Verbal fluency tests are considered to measure aspects of executive functioning such as organization, initiation and maintenance [37]. Participants were asked to generate as many words that begin with the letter "F" or names of animals in 60 seconds, respectively. Letter and semantic fluency scores were converted to z-scores using the entire sample's means and standard deviation at the initial cognitive assessment. A verbal fluency composite score was created by averaging the semantic and letter fluency z-scores at each visit. These cognitive measures have been validated for reliable administration over CATI [38, 39].

## 2.4. Covariates

Covariates included demographic, health behavior, and vascular risk variables collected during the baseline CATI or in-home visit. Demographic covariates included self-reported age, race, and sex, education level, yearly income, and region of residence. Participants reported their education level (< high school, high school degree, some college, $\geq$ college degree), and yearly income (<$20,000, $20,000–34,000, $35,000–74,000, $\geq$75,000 or unwilling to report). Region was categorized as residence in the stroke belt or non-stroke belt. Health behavior and vascular factors covariates included smoking status, alcohol use, exercise level, diabetes, hyperlipidemia, hypertension, and cardiovascular disease. Smoking status was categorized as never, former or current, and alcohol use as none, moderate ($\leq 4$ drinks/week for men, $\leq 3$ drinks/week for women) or heavy (> 4 drinks on any day or >14 drinks/week for men, > 3 drinks on any day or > 7 drinks/week for women). Exercise level was categorized as 4 or more times per week, 1 to 3 times per week, and none. Diabetes was defined as fasting glucose $\geq 126$ mg/dL, nonfasting glucose $\geq 200$ mg/dL, or self-reported use of diabetes medications. Dyslipidemia was defined as total cholesterol $\geq 6.22$ mmol/L (240 mg/dl), low density lipoproteins $\geq 4.14$ mmol/L (160 mg/dl), high density lipoproteins $\leq 1.04$ mmol/L (40 mg/dl) or use of self-reported anti-hyperlipidemic medications. Hypertension was defined as systolic blood pressure $\geq 140$ mm Hg or diastolic blood pressure $\geq 90$ mm Hg (average of two blood pressure measurements), or

self-reported use of hypertension medications. Body mass index (BMI) was calculated from height and weight measurements. BMI was computed as $kg/m^2$ and categorized according to Centers for Disease Control and Prevention guidelines: underweight ($<18.5$ $kg/m^2$), normal ($18.5–24.9$ $kg/m^2$), overweight ($25–29.9$ $kg/m^2$), or obese ($\geq 30$ $kg/m^2$). Prevalent cardiovascular disease was defined as self-reported coronary bypass, percutaneous coronary intervention, myocardial infarction, or myocardial infarction on electrocardiogram.

### 2.5. Statistical analyses

Descriptive statistics of the sample were calculated in SPSS 26. Longitudinal data were analyzed using latent growth curve models in Mplus version 7 [40] to determine the relationship of CRP to initial levels and rate of change in episodic memory and verbal fluency over time. Time was parametrized as years from baseline. In total, 5 visits (follow-up up to 12 years) were analyzed to maximize covariance coverage. We evaluated model fit by the Bayesian Information Criterion (BIC) [41]. Missing data were managed with full information maximum likelihood using all available data at each occasion. All models included CRP (logCRP or CRP > 90[th] percentile) as the primary predictor. Initially, separate latent growth curve models were estimated for each cognitive domain (episodic memory, verbal fluency) adjusting only for baseline age. Allowing linear versus curvilinear (age squared) change was compared in each of these models. To assess for evidence of practice effects, a spline modeling retest effects was included [42]. The best fitting models were retained for subsequent analyses that included evaluating the association of CRP and covariates. Compared with models allowing only linear change, fit was improved by allowing both linear and curvilinear change. Fit did not improve by modeling a spline for retest effects for each cognitive domain. Therefore, models estimating retest effects were not used. Subsequently, we built adjusted models in three steps, Step 1 included all demographic covariates, Step 2 added the health behavior covariates, and Step 3 added vascular risk factors. While cardiovascular disease and vascular risk factors (i.e., hypertension, BMI) may be considered mediators between CRP and cognition, we included these covariates in our final models to determine if there was an impact of CRP even after accounting for cardiovascular disease and vascular risk factors. The covariate-adjusted models fit the data well as noted by the improved model fit for each cognitive trajectory as demonstrated by smaller BICs (episodic memory BICs 127001.522, 103072.774; verbal fluency BICs 137901.434, 111822.316).

Multiple-group modeling was used to compare the magnitude of associations between CRP and cognitive trajectory between Black and White people in the unadjusted and adjusted models. Furthermore, given prior studies that report that the association between CRP and cognitive decline is stronger in midlife [19] we conducted post-hoc multiple-group modeling to evaluate whether the association between CRP and cognitive trajectory differed by age groups (midlife < 65 years of age vs late life $\geq$ 65).

## 3. Results

Table 1 shows the demographic and health characteristics of the overall sample and by race. Black participants were younger, had lower income, fewer years of education, were more likely to be overweight/obese, and were more likely to have hypertension and diabetes compared to White participants. White participants were more likely to be male and have cardiovascular disease and dyslipidemia. Black participants had higher median CRP than White participants.

### 3.1. Cognitive trajectories

Table 2 shows the unadjusted and adjusted models for the association of logCRP with episodic memory and verbal fluency. For memory and verbal fluency, higher CRP was associated with

**Table 1. Baseline characteristics by race.**

| | All (N = 21,782) | Black (n = 7,974) | White (n = 13,808) | p-value |
|---|---|---|---|---|
| **Demographics** | | | | |
| Male % (n) | 44.0 (9,583) | 36.0 (2,873) | 48.6 (6,710) | <0.001 |
| Stroke Belt % (n) | 55.8 (12,155) | 51.0 (4,069) | 58.6 (8,086) | <0.001 |
| Age, mean (SD) | 64.1 (9.1) | 63.1 (8.8) | 64.7 (9.2) | <0.001 |
| Education* | N = 21,773 | n = 7,970 | n = 13,803 | <0.001 |
| < HS, % (n) | 9.5 (2,072) | 15.6 (1,241) | 6.0 (831) | |
| HS grad % (n) | 25.0 (5,454) | 27.2 (2,165) | 23.8 (3,289) | |
| Some college % (n) | 27.3 (5,944) | 28.1 (2,243) | 26.8 (3,701) | |
| ≥ College grad % (n) | 38.1 (8,303) | 29.1 (2,321) | 43.3 (5,982) | |
| Income, $1,000/year, % (n) | | | | <0.001 |
| < 20 | 14.8 (3,233) | 22.9 (1,824) | 10.2 (1,409) | |
| 20–34 | 23.3 (5,077) | 26.2 (2,088) | 21.6 (2,989) | |
| 35–74 | 32.1 (6,998) | 29.0 (2,316) | 33.9 (4,682) | |
| ≥75 | 18.1 (3,945) | 10.8 (860) | 22.3 (3085) | |
| Refused | 11.6 (2,529) | 11.1 (886) | 11.9 (1,643) | |
| **Health Behaviors** | | | | |
| Alcohol amount % (n)* | N = 21,394 | n = 7,785 | n = 13,609 | <0.001 |
| None | 60.3 (12,911) | 70.1 (5,457) | 54.8 (7,454) | |
| Moderate | 35.5 (7,592) | 27.5 (2,137) | 40.1 (5,455) | |
| Heavy | 4.2 (891) | 2.5 (191) | 5.1 (700) | |
| Smoking frequency % (n)* | N = 21,701 | n = 7,939 | n = 13,762 | <0.001 |
| Never/Past | 86.7 (18,819) | 84.1 (6,677) | 88.2 (12,142) | |
| Current | 13.3 (2,882) | 15.9 (1,262) | 11.8 (1,620) | |
| No weekly exercise % (n)* | 32.0 (6,872) | 35.0 (2,756) | 30.2 (4,116) | <0.001 |
| | N = 21,488 | n = 7,871 | n = 13,617 | |
| **Vascular Risk Factors** | | | | |
| Diabetes % (n)* | 18.7 (4,043) | 26.7 (2,120) | 14.0 (1,923) | <0.001 |
| | N = 21,649 | n = 7,927 | n = 13,722 | |
| Prevalent CVD % (n)* | 15.7 (3,368) | 13.0 (1,019) | 17.3 (2,349) | <0.001 |
| | N = 21,436 | n = 7,834 | n = 13,602 | |
| Dyslipidemia % (n)* | 57.9 (12,485) | 53.4 (4,212) | 60.5 (8,273) | <0.001 |
| | N = 21,559 | n = 7,888 | n = 13,671 | |
| Hypertension % (n)* | 71.9 (15,625) | 83.1 (6,619) | 65.4 (9,006) | <0.001 |
| | N = 21,736 | n = 7,964 | n = 13,772 | |
| BMI Category % (n)* | N = 21,653 | n = 7,913 | n = 13,740 | <0.001 |
| Underweight/Normal | 24.6 (5,327) | 16.7 (1,323) | 29.1 (4,004) | |
| Overweight/Obese | 75.4 (16,326) | 83.3 (6,590) | 70.9 (9,736) | |
| CRP mg/L, median (IQR) | 2.1 (0.9–4.8) | 2.8 (1.2–6.3) | 1.8 (0.8–4.1) | <0.001 |

*Note.* CVD = cardiovascular disease, BMI = body mass index, CRP = C-reactive protein.

worse initial scores but not with change over time. After adjusting for covariates (Step 3), there was no independent association of CRP with memory, whereas the association of CRP with baseline verbal fluency remained. Similar patterns were observed when evaluating the association of elevated CRP (>90[th] percentile; (Table 2) on cognitive trajectories. Elevated CRP was associated with worse initial memory and verbal fluency scores but did not influence slope. After adjusting for covariates, the association between elevated CRP and baseline cognition

**Table 2. Associations of CRP with memory and verbal fluency trajectories.**

| | logCRP | | | | | | | |
|---|---|---|---|---|---|---|---|---|
| | **Unadjusted Model** | | **Adjusted Model 1** | | **Adjusted Model 2** | | **Adjusted Model 3** | |
| | Initial Level | Slope | Initial Level | Slope | Initial Level | Slope | Initial Level | Slope |
| Memory | -.039(.013)** | .001(.002) | -.036(.012)** | .001(.002) | -.027(.013)* | .002(.002) | -.005(.013) | .001(.002) |
| Fluency | -.195(.013)*** | .004(.002)* | -.077(.012)*** | .002(.002) | -.061(.012)*** | .002(.002) | -.042(.013)** | .003(.002) |
| | **90th Percentile CRP** | | | | | | | |
| | **Unadjusted Model** | | **Adjusted Model 1** | | **Adjusted Model 2** | | **Adjusted Model 3** | |
| | Initial Level | Slope | Initial Level | Slope | Initial Level | Slope | Initial Level | Slope |
| Memory | -.026(.022) | .001(.003) | -.077(.021)*** | -.001(.004) | -.066(.021)** | -.001(.004) | -.045(.022)* | -.001(.004) |
| Fluency | -.132(.021)*** | -.001(.003) | -.089(.020)*** | -.003(.003) | -071(.020)*** | -.003(.003) | -.052(.021)* | -.003(.003) |

*Note*. Values reflect unstandardized parameter estimates (standard error); adjusted model 1 includes all demographic covariates; adjusted model 2 includes the health behavior covariates; adjusted model 3 added vascular risk factors;

*p < .05,

**p < .01,

***p < .001. CRP = C reactive protein.

was partly attenuated, although it remained statistically significant. Multiple group models (Tables 3 and 4) revealed that the association between CRP (both continuous and 90th percentile status) and cognitive trajectories did not differ reliably by race (all *p*s > .05).

Post-hoc multiple group models by age group revealed distinct associations between CRP and cognitive trajectories (Table 5). For memory, higher CRP was associated with worse initial scores among participants in midlife compared with older participants, and this interaction remained after adjusting for covariates. There was no association of CRP on memory slope and this did not differ by age group (all *p*s > .05). For verbal fluency, higher CRP was associated with worse initial scores and this did not differ by age group. There was a counterintuitive association between CRP and change in fluency scores over time such that among older adults

**Table 3. Multiple group comparisons of associations of logCRP with memory and verbal fluency by race.**

| | Memory | | | | | | | |
|---|---|---|---|---|---|---|---|---|
| | **Unadjusted Model** | | **Adjusted Model 1** | | **Adjusted Model 2** | | **Adjusted Model 3** | |
| | Intercept | Slope | Intercept | Slope | Intercept | Slope | Intercept | Slope |
| White | -.005(.016) | .000(.002) | -.030(.015) | .001(.003) | -.019(.016) | .001(.003) | .003(.016) | .001(.003) |
| Black | .020(.022) | .002(.003) | -.047(.021)* | .001(.004) | -.042(.022) | .002(.004) | -.021(.023) | .001(.004) |
| White vs Black | -.025(.028) | -.002(.004) | .017(.026) | .000(.004) | .023(.027) | -.001(.005) | .024(.028) | .000(.005) |
| | **Verbal Fluency** | | | | | | | |
| | **Unadjusted Model** | | **Adjusted Model 1** | | **Adjusted Model 2** | | **Adjusted Model 3** | |
| | Intercept | Slope | Intercept | Slope | Intercept | Slope | Intercept | Slope |
| White | -.155(.016)*** | .002(.002) | -.090(.016)*** | .001(.002) | -.072(.016)*** | .002(.002) | -.055(.017)** | .003(.003) |
| Black | -.086(.019)*** | .004(.003) | -.049(.019)** | .004(.003) | -.037(.019) | .004(.003) | -.019(.020) | .005(.003) |
| White vs Black | -.070(.025)** | -.002(.003) | -.041(.024) | -.003(.004) | -.035(.025) | -.002(.004) | -.036(.026) | -.002(.004) |

*Note*. Values reflect unstandardized parameter estimates (standard error); adjusted model 1 includes all demographic covariates; adjusted model 2 includes the health behavior covariates; adjusted model 3 added vascular risk factors;

*p < .05,

**p < .01,

***p < .001. CRP = C reactive protein.

**Table 4. Multiple group comparisons of associations of CRP above the 90$^{th}$ percentile with memory and language by race.**

| | Memory | | | | | | | |
|---|---|---|---|---|---|---|---|---|
| | Unadjusted Model | | Adjusted Model 1 | | Adjusted Model 2 | | Adjusted Model 3 | |
| | Intercept | Slope | Intercept | Slope | Intercept | Slope | Intercept | Slope |
| White | -.034(.027) | .001(.004) | -.070(.025)** | -.003(.004) | -.058(.025)* | -.003(.004) | -.037(.026) | -.003(.005) |
| Black | -.009(.038) | .002(.006) | -.089(.036) | .002(.006) | -.081(.037) | .003(.006) | -.061(.038) | .003(.006) |
| White vs Black | -.026(.047) | -.001(.007) | .019(.044) | -.005(.008) | .023(.045) | -.006(.008) | .024(.046) | -.006(.008) |
| | Language | | | | | | | |
| | Unadjusted Model | | Adjusted Model 1 | | Adjusted Model 2 | | Adjusted Model 3 | |
| | Intercept | Slope | Intercept | Slope | Intercept | Slope | Intercept | Slope |
| White | -.148(.026)*** | -.002(.004) | -.095(.026)*** | -.005(.004) | -.075(.026) | -.004(.004) | -.061(.027)* | -.004(.004) |
| Black | -.096(.033)** | .000(.004) | -.071(.032)* | -.001(.004) | -.055(.032) | -.002(.004) | -.029(.032) | -.003(.005) |
| White vs Black | -.052(.042) | -.002(.006) | -.024(.041) | -.004(.006) | -.020(.041) | -.002(.006) | -.033(.042) | .000(.006) |

*Note.* Values reflect unstandardized parameter estimates (standard error); adjusted model 1 includes all demographic covariates; adjusted model 2 includes the health behavior covariates; adjusted model 3 added vascular risk factors;

*p < .05,

**p < .01,

***p < .001. CRP = C reactive protein.

higher CRP was associated with greater stability of trajectories, whereas there was no association between CRP and slope among participants in midlife (*p* = .02). The association of CRP with verbal fluency slope by age group was attenuated after adjusting for covariates. None of the previously mentioned results remained when evaluating the association of elevated CRP (>90$^{th}$ percentile; Table 6) on cognitive trajectories.

## 4. Discussion

In this nationally representative, population-based sample of White and Black American adults aged ≥45 years at baseline, higher CRP was associated with worse memory and verbal fluency

**Table 5. Multiple group comparisons of associations of logCRP with memory and verbal fluency by age.**

| | Memory | | | | | |
|---|---|---|---|---|---|---|
| | Adjusted Model 1 | | Adjusted Model 2 | | Adjusted Model 3 | |
| | Intercept | Slope | Intercept | Slope | Intercept | Slope |
| Midlife | -.066(.016)*** | -.002(.003) | -.055(.016)** | -.001(.003) | -.031(.017) | -.001(.003) |
| Late life | .002(.020) | .005(.004) | .006(.020) | .006(.004) | -.023(.021) | .005(.004) |
| Midlife vs Late life | -.068(.026)** | -.006(.005) | -.062(.026)* | -.007(.005) | -.055(.027)* | -.006(.005) |
| | Verbal Fluency | | | | | |
| | Adjusted Model 1 | | Adjusted Model 2 | | Adjusted Model 3 | |
| | Intercept | Slope | Intercept | Slope | Intercept | Slope |
| Midlife | -.085(.016)*** | -.001(.002) | -.067(.017)*** | -.001(.002) | -.043(.018)* | .002(.002) |
| Late life | -.072(.018)*** | .007(.003)* | -.061(.018)** | .008(.003)** | -.050(.018)** | .008(.003)** |
| Midlife vs Late life | -.014(.024) | -.008(.004)* | -.006(.025) | -.008(.004)* | -.007(.026) | -.007(.004) |

*Note.* Values reflect unstandardized parameter estimates (standard error); adjusted model 1 includes all demographic covariates; adjusted model 2 includes the health behavior covariates; adjusted model 3 added vascular risk factors;

*p < .05,

**p < .01,

***p < .001. CRP = C reactive protein.

**Table 6. Multiple group comparisons of associations of CRP above the 90th percentile with memory and verbal fluency by age.**

| | Memory | | | | | |
| --- | --- | --- | --- | --- | --- | --- |
| | Adjusted Model 1 | | Adjusted Model 2 | | Adjusted Model 3 | |
| | Intercept | Slope | Intercept | Slope | Intercept | Slope |
| Midlife | -.102(.027)*** | -.001(.004) | -.091(.026)** | -.001(.004) | -.068(.028)* | .000(.005) |
| Late life | -.043(.033) | .000(.006) | -.033(.034) | .000(.006) | -.015(.034) | -.001(.007) |
| Midlife vs Late life | -.058(.043) | -.001(.008) | -.058(.043) | .000(.008) | -.053(.044) | .001(.008) |
| | Verbal Fluency | | | | | |
| | Adjusted Model 1 | | Adjusted Model 2 | | Adjusted Model 3 | |
| | Intercept | Slope | Intercept | Slope | Intercept | Slope |
| Midlife | -.098(.027)*** | -.007(.004) | -.081(.028)** | -.007(.004) | -.055(.029) | -.005(.004) |
| Late life | -.081(.030)** | .003(.005) | -.064(.030)* | .003(.005) | -.056(.030) | .002(.005) |
| Midlife vs Late life | -.017(.040) | -.010(.006) | -.017(.041) | -.010(.006) | .001(.042) | -.007(.006) |

*Note*. Values reflect unstandardized parameter estimates (standard error); adjusted model 1 includes all demographic covariates; adjusted model 2 includes the health behavior covariates; adjusted model 3 added vascular risk factors;

*p < .05,

**p < .01,

***p < .001. CRP = C reactive protein.

at baseline but not with rate of cognitive decline over a span of 12 years. While the associations of continuous increments of CRP on baseline memory was attenuated after adjusting for demographic and vascular risk factors, the association remained significant among individuals with elevated CRP (>90th percentile). Higher CRP was more consistently associated with worse baseline verbal fluency with and without covariate adjustment. Furthermore, the association of CRP on cognition did not differ by race.

Our large cohort of Black and White adults followed longitudinally allowed us to assess the influence of CRP on cognitive trajectories. While the association of CRP on baseline cognition was robust, our hypothesis that elevated CRP would increase rate of cognitive decline was not supported. There are a few possible explanations for this discrepancy, and our null findings compared to some literature. First, prior studies among older adults that reported an association between CRP and cognitive decline have largely used brief cognitive screeners (i.e., Mini-Mental State Examination) [43], defined cognitive decline based on a single follow-up [16], or had brief follow-up periods (≤ 10 years) [17]. In contrast, our longitudinal study, focused on discrete cognitive domains (memory, verbal fluency), used measures that were well validated for use in this cohort [38, 39], and leveraged up to 5 follow-ups over the span of 12 years to characterize rate of cognitive decline. Second, previous studies generally accounted for few potentially confounding demographic and health variables. For instance, most studies would only include basic demographics (i.e., age, sex/gender, education) [16, 43, 44], a few others controlled for relevant health behaviors (i.e., alcohol use, smoking) [7, 11] associated with elevated CRP, or medical comorbidities (i.e., cardiovascular disease) [6, 11, 19] that may lie in the casual pathway between inflammation and cognitive decline. Third, we used race-specific cut-offs to determine elevated CRP, which may have influenced results. However, given the variability in CRP concentrations due to genetic ancestry, we believe our approach allows for robust evaluation of CRP across racial/ethnic groups. Moreover, the association between CRP measured continuously on rate of decline did not differ substantially from the associations using cut-offs, which suggests that a different or lower cut-off level for CRP would yield similar null results. As such, given these methodological differences, it may be possible that concentrations of CRP only affect cognitive level and not rate

of decline among older adults. Fourth, another possible explanation was that the effect of CRP on rate of decline may take place before the age of 65, given various studies suggesting that age-related cognitive decline accelerates during midlife [45], and that higher levels of CRP during midlife are associated with a steeper rate of cognitive decline [19], white matter disease [46, 47], and higher risk of dementia in late-life [6]. While we did not find a faster rate of cognitive decline among participants in midlife in our post-hoc analyses, we did find that among older adults higher CRP was associated with greater stability in verbal fluency trajectories (less decline). Prior studies have reported this inverse relationship between CRP and cognition among older adults, which may be due to successful cognitive aging and survival bias [48, 49]. Differences in mid- and late life associations were only observed when evaluating CRP continuously but largely attenuated when adjusting for covariates and not present when evaluating individuals with elevated CRP (>90[th] percentile). Lastly, while we evaluated change in cognition over up to 12 years, this may not be sufficient time to detect cognitive decline, even given our large sample size. For instance, a recent study found an association between midlife CRP and 20-year cognitive decline that was robust to adjustment for multiple covariates [19].

The availability of cognitive data in two domains (episodic memory and verbal fluency) allowed for investigation in domain-specific associations of CRP. Our results indicate that verbal fluency may be more susceptible to the effects of inflammation than episodic memory. Verbal fluency involves skills related to executive functioning (i.e., updating, inhibition) [37, 50] and is associated with integrity of frontal cortical structures, for letter fluency, and frontal and temporal-parietal regions, for semantic fluency [51]. As CRP is considered a marker of vascular disease, these results are in line with research that links vascular risk factors and markers of cerebrovascular disease with impairments in executive functioning [52, 53]. Similarly, a recent study reported that among older adults, higher baseline CRP was associated with reduced blood flow in frontal regions and the anterior cingulate cortex which are both regions associated with executive functioning [54]. Inflammation may lead to impaired endothelial functioning which is associated with white matter hyperintensities [55, 56]. In contrast, while studies have reported an association between elevated levels of CRP and worse performance and declines in episodic memory [19, 57, 58], we only found a robust association with worse memory at CRP levels above the 90[th] percentile. Inflammation may impact cognition through different mechanisms; it may lead to impaired endothelial functioning which is associated with white matter hyperintensities [55, 56], and with smaller brain volume [59]. Structures important for memory, such as the hippocampus, have high concentrations of pro-inflammatory cytokines and receptors which may be vulnerable to systemic inflammation. Given these domain-specific associations even after adjusting for vascular risk factors and cardiovascular disease, there may be different neurologic pathways through which inflammation may impact cognition.

We did not find that race modified the association between inflammation and cognitive decline. Given prior evidence for race differences in peripheral immune function [25, 26] and studies that have found a stronger association between markers of cerebrovascular disease (i.e., white matter hyperintensities) [53] and metabolic disorders (i.e., diabetes) [60] with cognition among Black compared to White people, we expected to find differences in the relationship between inflammation and cognition by race in the current study. In fact, a recent study from the Health Aging in Neighborhoods of Diversity across the Life Span (HANDLS) study found an association between higher CRP and worse cognitive performance among Black compared to White Americans [61]. However, participants in the HANDLS study were on average in midlife at baseline whereas our sample was older, HANDLS only included 2 cognitive assessments, and the association of CRP with cognition was only found on a test of attention but not on measures of episodic memory or verbal fluency, which mirrors our findings. Moreover, our results are in line with a recent study that did not find race differences in the association

between midlife levels of CRP and cognitive decline in the Atherosclerosis Risk in Communities (ARIC) Study [19]. While the ARIC Study evaluated CRP $\geq$ 3 mg/L [32, 33], we expanded on these results by additionally evaluating CRP using race-specific higher cutoffs ($\geq$ 8 mg/L for White adults and $\geq$ 12.3 mg/L for Black adults) given that CRP levels differ by race due to genetic ancestry [22]. Taken together, most results suggest that among older adults, the association of CRP with cognitive trajectories does not differ by race.

There are several strengths and weaknesses of this study. Strengths include the use of a large, national biracial cohort with serial cognitive assessments in two domains for up to 12 years, the use of structural equation modeling to estimate latent growth curve cognitive trajectories, and the inclusion of potentially confounding demographic and medical factors. A limitation is that it is unknown if findings are generalizable to individuals of other racial/ethnic backgrounds, given that our cohort only included White and Black adults. Another limitation was that given the resource limitations of a large-scale national study, we relied on telephone-based cognitive assessments. Although prior studies have validated telephone administration of cognitive measures and reported that they are comparable to face-to-face evaluations [38, 62], future studies should include a more extensive cognitive battery and in-person evaluations in order to improve sensitivity to cognitive decline. Another limitation is that we measured CRP only once. CRP level can increase in response to injury, infection, and inflammation [9, 63]. Although several studies indicate that CRP >3 mg/L indicates low grade-inflammation [64], and that CRP plays an important role in systemic inflammation [9], studies have shown that CRP varies within individuals over time and that at least three measurements are required to reliably establish the true mean [65]. It is possible that some participants had acute illness at the time of the CRP measurement, which would bias results towards the null hypothesis. However, the CRP distribution in our cohort was in line with what is expected based on other studies in the United States [66]. In addition, an increase in CRP over time has been associated with cognitive decline in two studies [67, 68], so longitudinal measurement may further help clarify the role of CRP on risk of cognitive impairment. Another similar potential limitation is that we only measured several covariates (i.e., height and weight) at baseline. Like most epidemiologic studies, for reasons of pragmatism, all baseline variables cannot be measured twice, which would improve precision of risk estimates. The impact of this limitation would be a bias to the null hypothesis. Lastly, we did not evaluate the association of CRP in conjunction with other inflammation biomarkers (i.e., IL-6). For instance, a summary index score of inflammation biomarkers including CRP, higher white blood cell counts, and lower serum albumin increased risk prediction of all-cause mortality, even after controlling for various demographic, sociocultural, and medical confounders [69]. Whether similar associations exist for rate of cognitive decline is unknown.

In conclusion, the current study with a large sample size (~20,000 adults) and well-characterized cohort in terms of demographics and medical factors, provides strong evidence that inflammation as measured with CRP at one time point is associated with cognitive level but may not increase rate of cognitive decline among older adults. Based on our findings, CRP may be used as a marker of cognitive impairment among older adults but may not be suitable for risk prediction for early cognitive decline. Further investigations are warranted to disentangle the association of this and other inflammatory markers on cognitive decline from the effects of other demographic and sociocultural risk factors.

## Acknowledgments

The authors thank the other investigators, staff, and participants of the REGARDS study for their valuable contributions. A full list of participating REGARDS investigators and institutions can be found at https://www.uab.edu/soph/regardsstudy/.

## Author Contributions

**Conceptualization:** Miguel Arce Rentería, Sarah R. Gillett, Leslie A. McClure, Jennifer J. Manly, Mary Cushman.

**Formal analysis:** Miguel Arce Rentería, Leslie A. McClure, Jennifer J. Manly.

**Funding acquisition:** Virginia G. Wadley, Virginia J. Howard, Jennifer J. Manly, Mary Cushman.

**Investigation:** Leslie A. McClure, Virginia G. Wadley, Virginia J. Howard, Mary Cushman.

**Methodology:** Miguel Arce Rentería, Virginia G. Wadley, Virginia J. Howard, Mary Cushman.

**Writing – original draft:** Miguel Arce Rentería, Sarah R. Gillett, Mary Cushman.

**Writing – review & editing:** Miguel Arce Rentería, Sarah R. Gillett, Leslie A. McClure, Virginia G. Wadley, Stephen P. Glasser, Virginia J. Howard, Brett M. Kissela, Frederick W. Unverzagt, Nancy S. Jenny, Jennifer J. Manly, Mary Cushman.

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
