## [Decision Letter · Decision Letter 0]

21 Aug 2020

PONE-D-20-23675

C-reactive protein and risk of cognitive decline: The REGARDS study

PLOS ONE

Dear Dr. Cushman,

Thank you for submitting your manuscript to PLOS ONE. After careful consideration by 2 Reviewers and an Academic Editor, all of the critiques of both Reviewers must be addressed in detail in a revision to determine publication status. If you are prepared to undertake the work required, I would be pleased to reconsider my decision, but revision of the original submission without directly addressing the critiques of the 2 Reviewers does not guarantee acceptance for publication in PLOS ONE. If the authors do not feel that the queries can be addressed, please consider submitting to another publication medium. A revised submission will be sent out for re-review. The authors are urged to have the manuscript given a hard copyedit for syntax and grammar.

**Comments to the Author**

1. Is the manuscript technically sound, and do the data support the conclusions?

Reviewer #1: Partly

Reviewer #2: No

2. Has the statistical analysis been performed appropriately and rigorously? 

Reviewer #1: Yes

Reviewer #2: Yes

3. Have the authors made all data underlying the findings in their manuscript fully available?

Reviewer #1: Yes

Reviewer #2: No

4. Is the manuscript presented in an intelligible fashion and written in standard English?

Reviewer #1: Yes

Reviewer #2: No

5. Review Comments to the Author

Reviewer #1: The association between CRP and cognitive function has been extensively studied. Yet, findings are conflicting and the current study has the advantage of a large national, biracial sample that was followed up for a relatively long duration.

I have several comments:

Abstract:

1. please add values to the reported results. It seems like subheadings were omitted in line with the journal formatting but the sentences were not revised accordingly (line 3- "To examine...", results should start with words such as "We found that..." (line 39). In addition, the sentence in line 43 ("Findings suggest...") is redundant.

Methods:

2. Why did the authors exclude participants who had one or more errors on the Six-Item Screener? This may limit external validity. Please explain.

3. Why was the 90th percentile chosen as CRP cutoff? Did the authors identify a threshold there? Please provide a rationale.

Results:

4. Line 203 "adjusting for covariates did not change the results". Although p-value remained significant, the effect sizes did become smaller. Please revise the sentence accordingly.

Discussion

Some important literature seems to be missing. For example, Boydoun et at (PMID 30356710) explored the association between CRP and cognitive decline in Whites and African Americans. In addition, there are multiple studies demonstrating an association between CRP and brain function and structure (e.g. PMID 29304217). These may help explaining the underlying mechanisms.

minor comments:

Line 109 "After fully adjustment..." please rephrase as other confounders may exist that the authors weren't aware of or were not available for this cohort.

Line 243: rate decline (change to "rate of decline")

Reviewer #2: The manuscript “C-reactive protein and risk of cognitive decline: The REGARDS study” concerns an interesting theme related to aging. However, some alterations are needed in order to improve the quality of the manuscript. English use and writing are acceptable, but could still be refined. For example, past and present tense are mixed throughout the text (past tense is preferable). The use of a single measure of CRP is an important limitation of the study, given that authors say several times that this would be a marker of “systemic inflammation”, what is not true. This should be carefully revised, as also the affirmation that “effects” were found (longitudinal analyses provided no significant findings, and cross-sectional associations cannot infer causality). Specific comments to each section are given below.

Introduction

1. The topic of the study is not only relevant to the population of the United States, but worldwide. By starting the introductory paragraph of the manuscript specifically mentioning the United States, the authors seem to restrict their own interest to this country. I suggest editing the sentence to make it more general.

2. One important point about the relationship between inflammation and cognitive decline is that, mainly, it has been shown within the status of low-grade inflammation, which may have very different metabolic implications compared to acute inflammation. CRP may be used as a marker of low-grade inflammation, if repeated measures are used. This was not the case of the present study. So, in fact, authors have used a very general marker of non-specific inflammation to investigate such potential pathway affecting cognitive function. This has to be clearly stated in the manuscript since the beginning, and also recognized as a limitation of the study.

Methods

3. The topic “Design and procedures” should mention the maximum duration of follow-up.

4. Lines 83-84: Details about how the Stroke Belt is defined could be briefly presented. Again, authors need to consider that their manuscript will reach readers from many countries, not only from the USA.

5. Why classifying CRP as “low” or “elevated”? Naming those categories like this give the idea that both would be “not ideal”. If authors test the hypothesis that increased CRP would be associated with cognitive decline, then “elevated” (or “high”) could be one category, compared to “normal” values.

6. In the Methods, authors recognize that literature used a distinct cutoff for CRP, for what can be defined as “chronic systemic inflammation” (or low-grade inflammation). By using a very distinct cutoff based on other criteria (genetic ancestry affecting CRP in different races), authors in fact distance themselves from exploring low-grade inflammation (what would also demand repeated CRP measures, by the way). So, again, the fact that chronic systemic inflammation is not being investigated needs to be made very clear throughout the manuscript.

7. Lines 146-151: If fasting glucose, nonfasting glucose, total cholesterol, low density lipoproteins and high density lipoproteins were measured by the investigators, their analytical procedures should be reported, as done for CRP. Detailed information should be provided for blood pressure, weight and height assessments.

8. Were body weight and height also measured in duplicate as done for blood pressure? This information should be made clear. If they were not, this should also be reported as a limitation of the study.

9. Please check with the Editor if the Journal has any specific orientation about how to mention race / skin color categories. I suggest avoiding the use of the terms “blacks” and “whites”.

Results

10. Line 192. If categories of nutritional status based on BMI are given in the results, they should have been previously defined in the Methods.

11. Line 194. The word “greater” may suggest the idea of something that is positive, good. I suggest using “higher” instead of “greater” (here and all over the text) to avoid this misinterpretation.

Discussion

12. In the first paragraph, please indicate the population of the study (adults aged ≥45 years).

13. Line 221-223: Significant findings with baseline data do not infer causality, so the word “effect” should not be used. “Strong” could also be avoided. I suggest replacing “While the effect of continuous increments of CRP on baseline memory” by “While the associations between continuous increments of CRP and baseline memory”, and replacing “the effect remained strong” by “the association remained significant”.

14. Lines 227-228: Authors say that “the effect of CRP on baseline cognition was robust”, but cross-sectional associations did not test effect.

15. Lines 244-245: Avoid repeating “midlife” in the same sentence: “take place before the age of 65, in midlife given various studies suggesting that age-related cognitive accelerates during midlife”.

16. Authors did not discuss the fact that their single measure of CRP may have influenced the findings. As said before, a unique measure does not allow inferring the state of low-grade inflammation.

17. Another point that may have contributed to distinct findings compared to literature was the use of different cutoffs.

18. Line 260: Authors mention “systemic inflammation”, but again, they did not measured systemic inflammation. Please revise.

19. Lines 309-310: I do not agree with the conclusion, since authors again insist in saying that “systemic inflammation” was measured, but it was not. Also, the findings of the present study are not “strong evidence that systemic inflammation as measured with CRP has a deleterious effect on cognitive level”. Effect (and so, causality) is measured in longitudinal analysis, and your longitudinal analyses did not provide significant results to support this affirmation. The significant cross-sectional associations cannot infer any causal relationship, so authors should not call it “deleterious effect”.

20. Only 8 out of the 56 references of the manuscript are recent (published after 2016). I suggest including more recent publications about the topic.

Tables and figures

21. Table 1. I suggest including the “n” to all cells in which only the % is presented (categorical variables).

22. Table 1. Were all variables available to all 21,782 participants? If not, variables with missing values should present their exact “n”.

23. Table 1. The differences in baseline characteristics according to race should be given (either if there were, or in the footnote in case there were not).

6. PLOS authors have the option to publish the peer review history of their article (what does this mean?). If published, this will include your full peer review and any attached files.

**Do you want your identity to be public for this peer review?** For information about this choice, including consent withdrawal, please see our Privacy Policy.

Reviewer #1: No

Reviewer #2: No

We look forward to receiving your revised manuscript.

Kind regards,

Stephen D. Ginsberg, Ph.D.

Section Editor

PLOS ONE

"REGARDS was approved by Institutional Review Boards of all participating institutions and all participants provided written informed consent. Interviewers were trained to identify participants answering questions in a manner suggesting lack of comprehension, and such participants were not included further. Potential participants who were able to respond to telephone questions provided verbal consent, which was followed by written consent at an in home visit.".   

---

## [Author Response · Author response to Decision Letter 0]

13 Nov 2020

Dear Dr. Ginsberg, 

We appreciate the careful review of our manuscript and are delighted that you will consider a revised version for publication. We have included a tracked changes version of the manuscript. We hope that our changes have adequately addressed the suggestions and that the revision will be suitable for publication in PLOS ONE. 

Reviewer #1: The association between CRP and cognitive function has been extensively studied. Yet, findings are conflicting, and the current study has the advantage of a large national, biracial sample that was followed up for a relatively long duration. I have several comments:

1. Abstract: 

Please add values to the reported results. It seems like subheadings were omitted in line with the journal formatting, but the sentences were not revised accordingly (line 3- "To examine...", results should start with words such as "We found that..." (line 39). In addition, the sentence in line 43 ("Findings suggest...") is redundant. 

We have added values throughout the abstract and made the recommendation changes to the text. 

2. Methods: 

Why did the authors exclude participants who had one or more errors on the Six-Item Screener? This may limit external validity. Please explain.

In order to exclude people with baseline impairment to allow study of potential causal relationships, we excluded participants with a score of 4 or fewer at baseline, which would indicate 2 or more errors on the Six-Item Screener (SIS). This cut-off is largely cited as indicating severe cognitive impairment and excluded 6.7% of participants. An indication of a participant experiencing severe cognitive impairment at baseline might also suggest other underlying medical factors (i.e., a neurodegenerative disease) which may in turn limit our ability to detect an association between CRP and cognition. Given our aim is to evaluate whether CRP would be helpful in predicting cognitive decline as to identify individuals at higher risk of cognitive decline and dementia, excluding participants with cognitive impairment at baseline does not limit our external validity. We updated the text in the methods section to clarify this. 

3. Why was the 90th percentile chosen as CRP cutoff? Did the authors identify a threshold there? Please provide a rationale.

We chose a race-specific 90th percentile cut-off to determine “elevated” CRP given recent work that differences in CRP concentration among racially/ethnically diverse populations may be partially driven by genetic ancestry, suggesting that a single threshold value of CRP may not be appropriate. We chose the 90th percentile to select the highest possible range in given racial group. In addition, in another REGARDS report, there was no association of the typically studied CRP cutoff of 3 mg/L with stroke in Black people in REGARDS. This rationale has been clarified in the methods section. 

4. Results: 

Line 203 "adjusting for covariates did not change the results". Although p-value remained significant, the effect sizes did become smaller. Please revise the sentence accordingly.

 We did not intend to be misleading and updated the text accordingly. 

5. Discussion:

Some important literature seems to be missing. For example, Boydoun et at (PMID 30356710) explored the association between CRP and cognitive decline in Whites and African Americans. In addition, there are multiple studies demonstrating an association between CRP and brain function and structure (e.g. PMID 29304217). These may help explaining the underlying mechanisms.

We have included these articles into our discussion, specifically relating how the Boydoun study differs from ours. We also included the study of CRP and blood brain flow as further evidence of potential mechanisms through which CRP impacts cognitive functioning. 

6. Minor comments:

Line 109 "After fully adjustment..." please rephrase as other confounders may exist that the authors weren't aware of or were not available for this cohort.

 We have removed “fully” from that and similar sentences. 

7. Line 243: rate decline (change to "rate of decline")

We have updated the text accordingly. 

Reviewer #2:

The manuscript “C-reactive protein and risk of cognitive decline: The REGARDS study” concerns an interesting theme related to aging. However, some alterations are needed in order to improve the quality of the manuscript. English use and writing are acceptable, but could still be refined. For example, past and present tense are mixed throughout the text (past tense is preferable). The use of a single measure of CRP is an important limitation of the study, given that authors say several times that this would be a marker of “systemic inflammation”, what is not true. This should be carefully revised, as also the affirmation that “effects” were found (longitudinal analyses provided no significant findings, and cross-sectional associations cannot infer causality). 

Specific comments to each section are given below.

Introduction

8. The topic of the study is not only relevant to the population of the United States, but worldwide. By starting the introductory paragraph of the manuscript specifically mentioning the United States, the authors seem to restrict their own interest to this country. I suggest editing the sentence to make it more general.

We updated the text accordingly to reflect the increase in global population aging. 

9. One important point about the relationship between inflammation and cognitive decline is that, mainly, it has been shown within the status of low-grade inflammation, which may have very different metabolic implications compared to acute inflammation. CRP may be used as a marker of low-grade inflammation, if repeated measures are used. This was not the case of the present study. So, in fact, authors have used a very general marker of non-specific inflammation to investigate such potential pathway affecting cognitive function. This has to be clearly stated in the manuscript since the beginning, and also recognized as a limitation of the study.

While we agree that multiple measurements of CRP over time would be ideal to establish chronic inflammation, nearly all articles on CRP as a risk factor for chronic diseases in initially healthy populations, including outcomes like mortality, heart disease, stroke, and cognitive impairment, measured CRP only once. The first commercially used assay for CRP for cardiovascular risk prediction (which was used in this study) was developed using the REGARDS central laboratory-developed ELISA assay as a gold standard in the 1990’s, and use of this test for cardiovascular risk prediction is supported in a number of guidelines, including the 2019 AHA/ACC guideline for primary prevention of cardiovascular disease (https://www.ahajournals.org/doi/pdf/10.1161/CIR.0000000000000677). The high sensitivity CRP assay also has reasonable within person variability as a marker of chronic low grade inflammation (Sakkinen et al., 1999 https://pubmed.ncbi.nlm.nih.gov/9927222/). It has been intensely studied in this context. We also previously showed that free-living people with levels of CRP in the range indicating acute inflammation (values >10 mg/L) are at increased cardiovascular risk (https://pubmed.ncbi.nlm.nih.gov/15983251/). Any participants who might have had acute illness around the time of the first visit would have spuriously high levels and their presence in the study would bias results to the null hypothesis. The CRP distributions in REGARDS were in line with what is expected from other studies, and participants would likely have rescheduled their baseline in-home visit if they were ill, so we do not expect there to be undue bias in this regard. To respond to the concerns raised, we have included additional information in our introduction regarding CRP as a marker of acute inflammation, and heavily edited the limitations paragraph regarding CRP as potential marker of non-specific inflammation and on the benefit of longitudinal CRP measurements. 

Methods

10. The topic “Design and procedures” should mention the maximum duration of follow-up.

We have indicated that maximum follow-up was up to 12 years in the methods. 

11. Lines 83-84: Details about how the Stroke Belt is defined could be briefly presented. Again, authors need to consider that their manuscript will reach readers from many countries, not only from the USA.

In the methods section, under design and procedures, we have included information indicating that a large southeastern region of the United States is referred to as the stroke belt given its well documented high rates of stroke observed in this region compared to the rest of the United States. 

12. Why classifying CRP as “low” or “elevated”? Naming those categories like this give the idea that both would be “not ideal”. If authors test the hypothesis that increased CRP would be associated with cognitive decline, then “elevated” (or “high”) could be one category, compared to “normal” values.

We have relabeled the CRP groups. Instead of using “low” and “elevated”, we use below and above the 90th percentile for our groupings in the methods section. However, for ease of readability, throughout the results we refer to above the 90th percentile as “elevated” CRP. We opted for not using the label of normal for values below the 90th percentile as they may still be above published cut-offs. 

13. In the Methods, authors recognize that literature used a distinct cutoff for CRP, for what can be defined as “chronic systemic inflammation” (or low-grade inflammation). By using a very distinct cutoff based on other criteria (genetic ancestry affecting CRP in different races), authors in fact distance themselves from exploring low-grade inflammation (what would also demand repeated CRP measures, by the way). So, again, the fact that chronic systemic inflammation is not being investigated needs to be made very clear throughout the manuscript.

We addressed these concerns in the reviewers previous comment. We agree that multiple CRP measurements would be ideal to determine chronic inflammation, but as with other epidemiological studies, it is unfeasible to have repeated CRP measurements. We do not believe that using race-specific cut-offs distances us from evaluating CRP as an indicator of systemic inflammation as several studies demonstrate that one-time assessment of CRP is associated with greater risk of worse health outcomes. As mentioned previously, we included in the limitations sections greater discussion regarding the role of CRP as a marker of inflammation and the need for longitudinal CRP assessment. To avoid confusion we removed the one instance of usage of the word “chronic” from the paper in favor of referring to levels of CRP relevant to disease risk prediction.

14. Lines 146-151: If fasting glucose, nonfasting glucose, total cholesterol, low density lipoproteins and high-density lipoproteins were measured by the investigators, their analytical procedures should be reported, as done for CRP. Detailed information should be provided for blood pressure, weight and height assessments.

We provided additional information regarding how these data were collected in the methods section. 

15. Were body weight and height also measured in duplicate as done for blood pressure? This information should be made clear. If they were not, this should also be reported as a limitation of the study.

Body weight and height were measured once during the in-home visit. We added this to the methods.

16. Please check with the Editor if the Journal has any specific orientation about how to mention race / skin color categories. I suggest avoiding the use of the terms “blacks” and “whites”.

We agree that these terms should be avoided, did not mean to infer we classified people by skin color (participants self-identified as reported), and changed all instances to Black people / participants and White people / participants. 

Results

17. Line 192. If categories of nutritional status based on BMI are given in the results, they should have been previously defined in the Methods.

We have included information regarding the BMI categories in the covariates section in the methods. 

18. Line 194. The word “greater” may suggest the idea of something that is positive, good. I suggest using “higher” instead of “greater” (here and all over the text) to avoid this misinterpretation.

We have updated the word greater for higher throughout the manuscript. 

Discussion.

19. In the first paragraph, please indicate the population of the study (adults aged ≥45 years).

We have included additional descriptive information regarding the population in the first paragraph of the discussion. 

20. Line 221-223: Significant findings with baseline data do not infer causality, so the word “effect” should not be used. “Strong” could also be avoided. I suggest replacing “While the effect of continuous increments of CRP on baseline memory” by “While the associations between continuous increments of CRP and baseline memory” and replacing “the effect remained strong” by “the association remained significant”. 

We have updated our text throughout the discussion to remove any words indicative of causality. 

21. Lines 227-228: Authors say that “the effect of CRP on baseline cognition was robust”, but cross-sectional associations did not test effect.

Similarly, we have removed the word “effect” for “association”. 

22. Lines 244-245: Avoid repeating “midlife” in the same sentence: “take place before the age of 65, in midlife given various studies suggesting that age-related cognitive accelerates during midlife”.

We removed the first instance of midlife in that sentence.

23. Authors did not discuss the fact that their single measure of CRP may have influenced the findings. As said before, a unique measure does not allow inferring the state of low-grade inflammation.

As mentioned previously, we have addressed this issue with greater information in the limitations paragraph of the discussion. 

24. Another point that may have contributed to distinct findings compared to literature was the use of different cutoffs.

We included this as a third point for the discrepancy in our findings from the literature in our second paragraph in the discussion, but the very null findings for CRP as a continuous variable argue against a missed association or that it represents a causal risk factor. 

25. Line 260: Authors mention “systemic inflammation”, but again, they did not measured systemic inflammation. Please revise.

We responded to this comment above. 

26. Lines 309-310: I do not agree with the conclusion, since authors again insist in saying that “systemic inflammation” was measured, but it was not. Also, the findings of the present study are not “strong evidence that systemic inflammation as measured with CRP has a deleterious effect on cognitive level”. Effect (and so, causality) is measured in longitudinal analysis, and your longitudinal analyses did not provide significant results to support this affirmation. The significant cross-sectional associations cannot infer any causal relationship, so authors should not call it “deleterious effect”.

We remove the term systemic and effect from our concluding paragraph. 

27. Only 8 out of the 56 references of the manuscript are recent (published after 2016). I suggest including more recent publications about the topic. 

We have included additional references throughout the document and now have 15 citations published on or after 2016. 

Tables and figures. 

28. Table 1. I suggest including the “n” to all cells in which only the % is presented (categorical variables).

To all categorical variables, we included now the n. 

29. Table 1. Were all variables available to all 21,782 participants? If not, variables with missing values should present their exact “n”.

We have indicated which variables had missing data with an asterisk and included the exact n in the table. 

30. Table 1. The differences in baseline characteristics according to race should be given (either if there were, or in the footnote in case there were not).

We had described the differences in baseline characteristics by racial group in the beginning of the results section, but we have now included a column that indicates the p-value to denote the differences we described in the text.

---

## [Decision Letter · Decision Letter 1]

3 Dec 2020

PONE-D-20-23675R1

C-reactive protein and risk of cognitive decline: The REGARDS study

PLOS ONE

Dear Dr. Cushman,

Thank you for resubmitting your work to PLOS ONE. Please make the corrections posed by Reviewer #2 so I can render a decision on this manuscript.

**Comments to the Author**

1. If the authors have adequately addressed your comments raised in a previous round of review and you feel that this manuscript is now acceptable for publication, you may indicate that here to bypass the “Comments to the Author” section, enter your conflict of interest statement in the “Confidential to Editor” section, and submit your "Accept" recommendation.

Reviewer #2: (No Response)

2. Is the manuscript technically sound, and do the data support the conclusions?

Reviewer #2: Yes

3. Has the statistical analysis been performed appropriately and rigorously? 

Reviewer #2: Yes

4. Have the authors made all data underlying the findings in their manuscript fully available?

Reviewer #2: No

5. Is the manuscript presented in an intelligible fashion and written in standard English?

Reviewer #2: Yes

6. Review Comments to the Author

Reviewer #2: The quality of the manuscript was improved. Please find some minor comments below.

Line 74. The word “in” appears twice. Please delete one.

Please include in the limitations of the study the fact that weight and height were not measured in duplicate.

There are still several sentences using “greater”, for which “higher” would be more appropriate.

There are still some sentences in which the word "effect" must be avoided, because of describing cross-sectional findings (example: line 241).

Line 267-269: Authors say that “recent studies have reported (…) greater risk of dementia in later life [6]”. However, reference 6 is not “recent”, since it was published in 2002.

7. PLOS authors have the option to publish the peer review history of their article (what does this mean?). If published, this will include your full peer review and any attached files.

**Do you want your identity to be public for this peer review?** For information about this choice, including consent withdrawal, please see our Privacy Policy.

Reviewer #2: No

We look forward to receiving your revised manuscript.

Kind regards,

Stephen D. Ginsberg, Ph.D.

Section Editor

PLOS ONE

---

## [Author Response · Author response to Decision Letter 1]

7 Dec 2020

Dear Dr. Ginsberg, 

We greatly appreciate the careful second review of our manuscript and are delighted that you will consider a revised version for publication. We have included a tracked changes version of the manuscript with changed text highlighted in yellow for convenience. We hope that our changes have adequately addressed the suggestions and that the revision will be suitable for publication in PLOS ONE. 

Reviewer #2: The quality of the manuscript was improved. Please find some minor comments below.

1. Line 74. The word “in” appears twice. Please delete one.

We have removed this typo.

2. Please include in the limitations of the study the fact that weight and height were not measured in duplicate.

While we agree it would be ideal to measure everything in duplicate, due to the inherent design of a large epidemiological study such as REGARDS, it is not feasible to accomplish that. We are not concerned that this lack of duplicate measurement for weight and height is a notable limitation to the aims of our particular study. While lack of a duplicate height and weight measurement may lead to measurement error in a confounder (BMI), we are able to overcome this potential measurement error due to the statistical power provided by our large sample size. Moreover, this potential issue would lead to BMI further attenuating the relationship of CRP with our cognitive outcomes thus biasing to the null hypothesis. Regardless, we have added the following statement in our limitations section:

“Another similar potential limitation is that we only measured several covariates (i.e., height and weight) at baseline. Like most epidemiologic studies, for reasons of pragmatism, all baseline variables cannot be measured twice, which would improve precision of risk estimates. The impact of this limitation would be a bias to the null hypothesis.”

3. There are still several sentences using “greater”, for which “higher” would be more appropriate.

We have removed the word “greater” throughout the manuscript and updated it for “higher” or “increase” where appropriate. 

4. There are still some sentences in which the word "effect" must be avoided, because of describing cross-sectional findings (example: line 241).

We have switched “effect” for “association” as appropriate throughout the manuscript. 

5. Line 267-269: Authors say that “recent studies have reported (…) greater risk of dementia in later life [6]”. However, reference 6 is not “recent”, since it was published in 2002.

We removed “recent” from that sentence.

Lastly, an additional comment to address was that Reviewer #2 answered “No” to the following question “Have the authors made all data underlying the findings in their manuscript fully available?” and we would just like to reiterate that we addressed this in our previous cover letter as suggested by editor with the following information:

“The data underlying the findings include potentially identifying participant information and cannot be made publicly available due to ethical/legal restrictions. However, data including statistical code from this manuscript are available to researchers who meet the criteria for access to confidential data. Data can be obtained upon request through the University of Alabama at Birmingham at regardsadmin@uab.edu.”

---

## [Editor Report · Decision Letter 2]

14 Dec 2020

C-reactive protein and risk of cognitive decline: The REGARDS study

PONE-D-20-23675R2

Dear Dr. Cushman,

We’re pleased to inform you that your manuscript has been judged scientifically suitable for publication and will be formally accepted for publication once it meets all outstanding technical requirements.

Kind regards,

Stephen D. Ginsberg, Ph.D.

Section Editor

PLOS ONE

---

## [Editor Report · Acceptance letter]

18 Dec 2020

PONE-D-20-23675R2 

C-reactive protein and risk of cognitive decline: The REGARDS study 

Dear Dr. Cushman:

I'm pleased to inform you that your manuscript has been deemed suitable for publication in PLOS ONE. Congratulations! Your manuscript is now with our production department. 

Kind regards, 

on behalf of

Dr. Stephen D. Ginsberg 

Section Editor

PLOS ONE